# Epigenetic Analysis of the Dopamine Transporter Gene *DAT1* with a Focus on Personality Traits in Athletes

**DOI:** 10.3390/ijms24108931

**Published:** 2023-05-18

**Authors:** Kinga Humińska-Lisowska, Krzysztof Chmielowiec, Aleksandra Strońska-Pluta, Jolanta Chmielowiec, Aleksandra Suchanecka, Jolanta Masiak, Monika Michałowska-Sawczyn, Agnieszka Boroń, Paweł Cięszczyk, Anna Grzywacz

**Affiliations:** 1Faculty of Physical Education, Gdansk University of Physical Education and Sport, K. Górskiego St. 1, 80-336 Gdansk, Poland; kinga.huminska-lisowska@awf.gda.pl (K.H.-L.); monikamichalowska@op.pl (M.M.-S.); cieszczyk@poczta.onet.pl (P.C.); 2Institute of Sports Sciences, The University of Physical Education in Krakow, 31-541 Kraków, Poland; 3Department of Hygiene and Epidemiology, Collegium Medicum, University of Zielona Góra, 28 Zyty St., 65-046 Zielona Góra, Poland; chmiele@vp.pl (K.C.); chmiele1@o2.pl (J.C.); 4Independent Laboratory of Health Promotion, Pomeranian Medical University in Szczecin, Powstańców Wielkopolskich 72 St., 70-111 Szczecin, Poland; aleksandra.stronska@pum.edu.pl (A.S.-P.); o.suchanecka@gmail.com (A.S.); 5Second Department of Psychiatry and Psychiatric Rehabilitation, Medical University of Lublin, 1 Głuska St., 20-059 Lublin, Poland; jolanta.masiak@umlub.pl; 6Department of Clinical and Molecular Biochemistry, Pomeranian Medical University in Szczecin, Aleja Powstańców Wielkopolskich 72 St., 70-111 Szczecin, Poland; agnieszka.boron@pum.edu.pl

**Keywords:** dopamine transporter, *DAT1*, methylation, athletes

## Abstract

Human phenotypes (traits) are determined by the selective use of a person’s unique genotype (DNA sequence), following exposure to environmental stimuli, such as exercise. Inducing profound changes in epigenetics may be an underlying factor of the beneficial effects of exercise. This study aimed to investigate the association between methylation in the promoter region of the *DAT1* gene and personality traits measured by the NEO-FFI questionnaire in a group of athletes. The study group included 163 athletes, and the control group consisted of 232 non-athletes. The obtained results show several significant differences between the studied groups of subjects. The Extraversion scale and the Conscientiousness scale results of the NEO-FFI are significantly higher in the group of athletes compared to controls. The total methylation and the number of methylated islands in the promoter region of the *DAT1* gene are higher in the study group. Pearson’s linear correlation between the total methylation, the number of methylated islands and the NEO-FFI shows significant results for the Extraversion and Agreeability scales. The total methylation and the number of methylated islands in the promoter region of the *DAT1* gene are higher in the study group. Pearson’s linear correlation between the total methylation, the number of methylated islands and the NEO-FFI shows significant results for the Extraversion and Agreeability scales. Our analysis of the methylation status of individual CpG sites revealed a new direction of research into the biological aspects of regulating dopamine release and personality traits in people practicing sports.

## 1. Introduction

Human phenotypes (traits) are determined by the selective use of a person’s unique genotype (DNA sequence) following exposure to environmental stimuli, such as exercise and childhood exposure to potentially negative experiences (such as marital conflict, parental psychopathological risk and poor-quality parent–infant relationships) [1,2,3,4,5]. When exercise stimuli are experimentally controlled, there are significant differences in how individuals respond, suggesting a significant influence of genetic factors [6]. Consequently, sport and exercise genomics has become a sub-discipline of molecular exercise physiology and aims to understand the organization and function of the genome in the sporting environment [7]. Sports genomics typically investigates whether common DNA sequence variants, such as single nucleotide polymorphisms (SNPs) [1], are associated with the phenotype of interest. Indeed, there are genomic differences between individuals that may explain differences in skeletal muscle fiber type [8], aerobic capacity [6] and muscle strength [9], and there are several specific genotypes that are associated with the attainment of elite athlete status [7,10]. Genetic approaches, such as linkage analysis, canonical gene studies, genome-wide association studies (GWAS) and whole exome or genome sequencing, can test the association between established genetic variation and measurable traits [11]. However, they are less able to explain and understand the role of the environment. Epigenetic regulation has been shown to play a key role in controlling and programming gene expression [12]. While heritable genetic factors contribute to approximately 50–60% of exercise-related traits [6,8,13,14,15], and the environment is responsible for the remainder, epigenetic mechanisms describe the integration of these two elements and allow the control of gene activity without changes in DNA sequence [12,16,17,18]. Thus, epigenetic regulation of genes in response to external stimuli is an important modulator of physiological adaptation, and interest in epigenetic research is of increasing interest in the sports and exercise field, as it can provide an understanding of how cells, tissues and organs adapt to exercise, training and related physiological challenges [19,20].

Life experiences, habits and our environment shape what and who we are by virtue of their impact on our epigenome and health. For instance, although identical twins share the same genome and are superficially phenotypically similar, they are unique individuals with definable differences. These differences result from distinct gene expression influenced by epigenetic factors. Behavior, nutrition and exposure to toxins and pollutants are among the lifestyle factors known to be associated with epigenetic modifications. For example, nutrition is a key environmental exposure from gestation to death that impacts our health by influencing epigenetic phenomena [21].

There are three interrelated categories of epigenetic mechanisms: DNA/RNA methylation, non-coding RNAs (ncRNAs) and post-translational modifications of histones. DNA methylation, the only known epigenetic mechanism that directly affects DNA, involves the addition of a methyl group to the 5th carbon of cytosine (5mC), the 6th carbon of adenine to form N6-methyladenine (6mA) and the 7th carbon of cytosine (5mC) and guanine to form 7-methylguanine (7mG). 5mC is the most common of these types of DNA methylation. The distribution of methylation largely determines whether gene expression is increased or decreased with methylation [22,23]. It has been suggested that methylation at the gene promoter or ‘enhancer’ site interferes with gene transcription; however, methylation at intergenic regions is generally thought to increase gene expression [23]. DNA methylation is relatively static in adults. However, there are two distinct phases of demethylation during gametogenesis and embryogenesis [24,25]. The exact mechanisms involved in reprogramming DNA methylation still need to be elucidated [26].

Inducing profound changes in epigenetics may be an underlying factor of the beneficial effects of exercise. The effects of exercise on total DNA methylation in various tissues, including skeletal muscle, adipose tissue and blood, have been demonstrated in many studies [26,27,28].

The physical determinants of success in sports are well-known in physical culture. They depend on somatic, motoric, technical and tactical preparation. Somatic predisposition is mainly genetically determined and verified at the selection stage, while the specificity of the discipline practiced determines physical preparation. Physical preparation is the result of a well-organized sports training program. However, most athletes fail to achieve mastery, despite their somatic predisposition and physical conditions. Therefore, there is an assumption that the psyche is behind the success of a physically well-prepared athlete. Thus, the psychological determinants of success in sports have become the focus of sports psychologists. Just as the physical determinants of success in sports are divided into four components, the mental determinants of eudaimonia during competition can include personality factors, temperamental factors, arousal control, self-confidence, mental toughness, concentration, relaxation and others. However, most attention has been given to personality [29,30,31,32].

The general profile of athletes in terms of the five-factor Big Five personality model [33] is low Neuroticism, high levels of Extraversion and Conscientiousness, and average levels of Openness to Experience and Agreeableness [34,35,36,37,38,39,40,41]. However, the profile is also largely influenced by the sport practiced, and the athlete’s personality state is dependent on this, making it extremely difficult to distinguish and define the most favorable personality type [42].

The *DAT1* gene, also identified as *SLC6A3* [43], is located on chromosome 5p15 and belongs to the thNa/Cl-dependent transporters family. This transporter is widely distributed in the brain in areas of dopaminergic activity. DAT is situated in the axonal terminals’ plasma membrane, receives dopamine from the synapse [37] and regulates dopamine levels in the extracellular space [44,45,46,47].

Dopamine activity has been experimentally linked to differences in personality traits using various methods [48]. However, associations between genes that regulate dopamine activity and global personality phenotypes have been less consistent [49]. This may be because of a significant vector of environmental factors (e.g., parental support, adverse life events, availability of resources), which also influences the development of personality traits, resulting in different phenotypes despite similar genotypes, depending on environmental conditions. This is commonly referred to as ‘phenotypic plasticity’ [50]. This means that different genotypes can respond differently to environmental factors. As a result, interactions with the environment can create a genotype pattern [51,52].

Thus, this study aimed to investigate the association between methylation in the promoter region of the *DAT1* gene and personality traits measured by the NEO-FFI questionnaire in a group of athletes.

## 2. Results

Differences in methylation levels at individual sites (islands) of the *DAT1* promoter were observed when analyzing the methylation status of individual CpG sites (Table 1, Figure 1). Of the 33 CpG sites, 18 showed significantly higher levels of methylation in athletes (sites 1,2,4,5,8,9,10,11,12,13,14,16,17,21,23,27,32 and 33). The highest percentage of methylation was recorded sequentially from the highest for the following islets in athletes: sites 14, 19, 22, 3, 15, 2, 18, 31, 1, 21, 24, 16, 33, 28, 12, 32, 4, 17, 20, 5, 9, 10, 13, 26, 27, 23, 25, 7, 29, 11, 6, 8 and 30; while in the control group the sites were 19, 22, 3, 14, 15, 18, 2, 24, 33, 28, 21, 16, 32, 1, 31, 20, 26, 8, 11, 13, 9, 10, 5, 25, 17, 4, 12, 29, 23, 27, 7, 30 and 6. When comparing the odds ratio of increased methylation in the athlete group compared to the control group, the sites 13; OR = 22.44, 14; OR = 8.23, 12; OR = 5.08, 4; OR = 4.94, 11; OR = 4.91, 1; OR = 3.74, 27; OR = 3.61, 17; OR = 3.56, 8; OR = 3.29 were significant (Table 1). 

The analysis of the total methylation of *DAT1* showed a statistically significant increase in the number of total methylated CpG islands in the athletes (56.70%) compared to the controls (42.85%) (Z = 7.125, *p* < 0.00001, Table 1).

While comparing the controls and the group of athletes, we observed significantly higher scores on the NEO Five-Factor Inventory Extraversion scale (M 6.84 vs. M 6.36, *p* = 0.0312) and the NEO Five-Factor Inventory Conscientiousness scale (M 7.19 vs. M 5.88, *p* < 0.00001). However, for the remaining three features of the NEO Five-Factor Inventory, such as the Neuroticism scale, Openness scale and Agreeability scale, no significant differences were found between the group of athletes and the control group (Table 2). 

Pearson’s linear correlation analysis in athletes showed a positive correlation between total methylation island levels and the NEO-FFI Extraversion scale (r = 0.174; *p* = 0.027) and the NEO-FFI Agreeability scale (r = 0.158; *p* = 0.044). In the control group, no significant Pearson’s linear correlation was observed in the five analyzed features of the NEO Five-Factor Inventory. For one trait, Pearson’s linear correlation approaches statistical significance, and an inverse negative correlation is observed between total methylation island levels and the NEO-FFI Extraversion scale (r = 0.118; *p* = 0.073, Table 3, Figure 1 and Figure 2).

## 3. Discussion

In this study, we analyzed the association between methylation in the *DAT1* promoter region and personality traits measured by the NEO-FFI in a group of athletes and controls. The obtained results show several significant differences between the studied groups of subjects. The Extraversion scale and the Conscientiousness scale results of the NEO-FFI are significantly higher in the group of athletes compared to controls. 

Interestingly, our results did not show any significant differences between the analyzed groups on the Neuroticism scale. In a study comparing athletes and champions, the latter had lower levels of Neuroticism and higher levels of Extraversion, Openness to Experience, Agreeableness and Conscientiousness. Neuroticism was an important personality determinant: the lower the level of Neuroticism, the more likely an athlete was to be classified as a champion. In each Big Five personality dimension, champions differed from other athletes [53].

Mirzaei et al [35] also attempted to investigate the relationship between personality traits measured by the Big Five and athletic performance [35]. The research sample included over 200 non-elite footballers and futsal soccer players. Of the personality traits, only Conscientiousness was significantly related to athletic performance—therefore, Conscientiousness alone was the only predictor of a athletic performance, according to the study’s authors.

In the same year, using the Big Five model, Steca et al. [54] studied over 800 athletes and non-athletes. In every dimension of the Big Five—except Neuroticism, where they scored lower—the most successful athletes in their discipline scored higher than the non-athletes. By contrast, less successful athletes only outperformed non-athletes on Extraversion and Agreeableness. More successful competitive athletes (champions) showed greater emotional stability (lower Neuroticism), Extraversion, Openness to Experience, Agreeableness and Conscientiousness than less effective athletes. In addition, the individual athletes were more energetic and open-minded than the team athletes. In another study, Piepiora and Witkowski [55] attempted to create psychological personality profiles of athletes competing in individual and team disciplines, depending on the pressure exerted on the opponent in the starting situation. Differences between sports disciplines in which pressure is exerted indirectly on the opponent, and those in which pressure is exerted directly on the opponent were found on the Neuroticism and Conscientiousness scales. The study groups, except volleyball and football players, differed on the neuroticism scale, while volleyball players were less likely to show Agreeableness and Conscientiousness than other athletes. As can be seen, the type of sport played has an impact on personality.

The results of a study focusing on the relationship between personality traits, derived from the Big Five model, and performance in the competitive electronic sports (eSports) video game League of Legends (LoL), suggest that players who reach higher levels in LoL tend to be less extroverted and less agreeable but more open to new experiences. This is one of the few reported links between eSports performance and personality traits. The field is still unknown but is developing rapidly [56].

In our study, the total methylation and the number of methylated islands in the promoter region of the *DAT1* gene are higher in the study group. Moreover, Pearson’s linear correlation between the total methylation, the number of methylated islands and the NEO-FFI shows significant results for the Extraversion and Agreeability scales.

Epigenetic markers, such as methylation of the *DAT1* gene promoter, may explain the underlying mechanisms of gene-environment interaction and how their interaction affects brain development [57,58,59]. The effects of gene methylation on gene expression are complex. However, gene methylation is generally considered a ‘silencing’ epigenetic mark. Various studies have found that methylation of CpG islands in the promoter region has an inhibitory effect on the initiation of transcription, resulting in a reduction in gene expression [60,61]. The observed link between the methylation of *DAT1* and its expression may reflect the high concentration of CpG islands in the gene, making *DAT1* expression particularly susceptible to modulation by epigenetic mechanisms, particularly DNA methylation [62].

Previous studies found that *DAT1* methylation derived from blood correlated with symptoms of hyperactivity and impulsivity in children and adolescents with ADHD [63] and with impulsivity (and basal ganglia DAT availability) in monkeys [64]; alcohol dependency and alcohol craving [65], nicotine [66] and cannabis [67] dependency in humans. Moreover, *DAT* methylation from peripheral blood showed a positive correlation with DAT availability of ventral striatum in healthy subjects [68].

Lifestyle interventions, including exercise and dietary supplementation, can modify DNA methylation and confer health benefits, but the underlying mechanisms are poorly understood. Hunter et al [69] show that acute exercise and dietary supplementation with omega-3 polyunsaturated fatty acids (n-3 PUFA) and extra virgin olive oil (EVOO) can induce changes in DNA methylation in leukocytes, possibly by modulating the expression of DNMT mRNA. Future studies are needed to elucidate lifestyle effects on DNA methylation.

It is known that the long-term regulation of gene expression is influenced by epigenetic mechanisms, such as DNA methylation, histone modifications and chromatin remodeling, which provide a molecular memory of the interactions between genes and the environment [70]. It can be hypothesized that hypermethylation of the *DAT* promoter inhibits transcription and, therefore, the expression of *DAT*, and results in elevated levels of dopamine (DA) in the synaptic cleft [71]. Accordingly, increased DAT availability should result in lower extracellular DA levels [72] and a lower signal-to-noise ratio, which, in turn, should allow the circuit to more easily reach the threshold for response. The increased levels of DAT in the GPi documented here may facilitate the initiation of responses to external (or internal) signals via the direct (GO) pathway, and the faster termination of ongoing reactions to facilitate the initiation of subsequent responses via the indirect (NO-GO) pathway. Essentially, Rajala et al [64] hypothesized that lower DA levels induced by higher DAT availability should facilitate reaching the response threshold and contribute to impulsive behavior.

The growing interest in underlying genetic and epigenetic mechanisms drives new sports and exercise science directions. Technological advances in molecular biology enable researchers to apply new techniques and generate new hypotheses more practically [20].

## 4. Materials and Methods

### 4.1. Participants

The study group included 163 athletes, and the control group consisted of 232 non-athletes. Table 4 shows the mean age of each group and the percentage of females and males. All participants were European, to reduce the possibility of genetic admixture and overcome potential population stratification problems.

The study was conducted according to the guidelines of the Helsinki Declaration and was approved by the Bioethics Committee of the Medical University of Pomerania in Szczecin. All participants were informed about the rules of the study, familiarized with the procedure and informed about the possibility of withdrawing from the study at any time.

None of the study participants was financially rewarded for participating in the project. The study was anonymized entirely, following the principles of personal data protection. Athletes were defined as people involved in sports on a professional level. The control group was selected based on age and sex. All procedures for comfort and concentration were carried out.

### 4.2. Assessment of the Methylation Status of the Dopamine Gene Transporter (DAT1) Promoter

DNA was isolated from peripheral blood using a DNA isolation kit (A&A Biotechnology, Gdynia, Poland). Bisulfite modifications were performed using the EZ DNA Methylation Kit (Zymo Research, Orange, CA, USA) on 250 ng DNA, according to the manufacturer’s instructions. A Mastercycler Epgradient S (Eppendorf, Germany) was used for the methylation-specific PCR assay.

Oligonucleotide primers were obtained from Genomed.pl (Warsaw, Poland) and designed using metprimer (http://www.urogene.org/cgi-bin/methprimer/methprimer.cgi, accessed on 29 April 2022). The status of the *DAT1* promoter (ENSG00000142319) was assessed by PCR using primers specific for the gene fragment, i.e., DATF: 5′-GGTTTTTGTTTTTTTTTTGTTGAG-3′; DATR: 5′-AAATCCCCTAAACCTAATCCC-3′. The PCR conditions for the amplification of the 447 bp fragment spanning the 33 CpG sites in the *DAT1* gene promoter are shown in Table 5.

The concentration of magnesium chloride ions was set at 2.5 mM. After amplification, PCR products were sequenced as previously described [73]. Briefly, the samples were verified by sequencing using the BigDye v3.1 kit (Applied Biosystems, Darmstadt, Germany). The samples were separated by ethanol extraction using an ABI Prism 3130XL (Applied Biosystems, Darmstadt, Germany) in a 36 cm POP7 polymeric capillary, using a reverse primer.

Sequencing chromatograms were analyzed using 4peaks software (v. 1.8., Mek & Tosj, Amsterdam, The Netherlands, https://nucleobytes.com/4peaks/index.html) (accessed on 29 April 2022). A G/A + G ratio of at least 20% of the total signal was considered positive for cytosine methylation. The formula for calculating the percentage of methylation in each subject was (G/(G + A) × 100).

### 4.3. Statistical Analysis

The Mann–Whitney U test was used to analyze and compare the total methylation level (%) of 33 CpG *DAT1* sites in the two groups of subjects. The personality traits of the athletes, as measured by the NEO Five-Factor Inventory, were compared with a control group using the same test.

The chi-squared test was used to analyze differences in the methylation percentage at individual CpG islands in the two groups of subjects, with *p* < 0.05 considered statistically significant. For these variables, the accepted significance level was 0.0015 (0.05/33), using the Bonferroni correction for multiple comparisons.

The relationship between the total methylation level (%) of 33 CpGs and the personality traits measured by NEO Five-Factor Inventory was shown separately in both study groups, using Pearson’s linear correlation.

All statistical analyses were performed using STATISTICA 13 (TIBCO Software, Inc., Palo Alto, CA, USA) and PQStat software (v. 1.8.2., Poznań, Poland).

## 5. Conclusions

The total methylation and the number of methylated islands in the promoter region of the *DAT1* gene are higher in the study group. Pearson’s linear correlation between the total methylation, the number of methylated islands and the NEO-FFI shows significant results for the Extraversion and Agreeability scales.

Our analysis of the methylation status of individual CpG sites revealed a new direction of research on the biological aspects of regulating dopamine release and personality traits in people practicing sports.

## Figures and Tables

**Figure 1 ijms-24-08931-f001:**
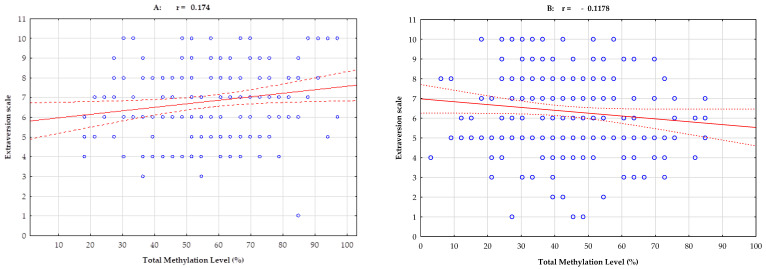
Pearson’s linear correlation between total methylation level, number of methylation islands and NEO-FFI Extraversion scale in a group of athletes (**A**) and controls (**B**). r—correlation coefficient. Blue dots are the recorded values for single measurements (several measurements may be in the same position). The red line indicates the determined linear correlation, while the red dotted lines indicate the range of the confidence interval from +95 to −95.

**Figure 2 ijms-24-08931-f002:**
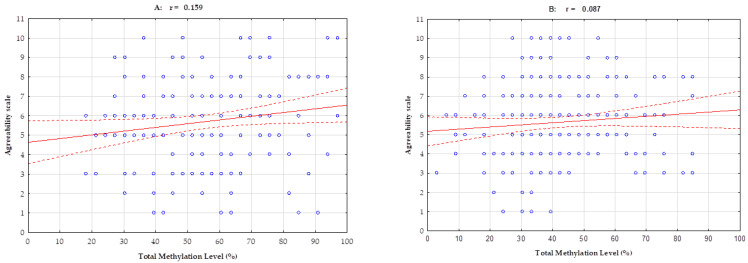
Pearson’s linear correlation between total methylation level, number of methylation islands and NEO-FFI Agreeability scale in a group of athletes (**A**) and controls (**B**). r—correlation coefficient. Blue dots are the recorded values for single measurements (several measurements may be in the same position). The red line indicates the determined linear correlation, while the red dotted lines indicate the range of the confidence interval from +95 to −95.

**Table 1 ijms-24-08931-t001:** Methylation status of 33 CpG *DAT1* sites in the athletes and controls.

CpG Site	Study Groups Methylation Level n (%)		Test χ^2^	*p* Value	OR	CL −95; +95
	Athletes n = 163	Controls n = 232				
1 #	139 85.28%	141 60.78%	27.845	0.0000	3.74 *	2.25; 6.21
2 #	148 90.80%	182 78.45%	10.620	0.0011	2.71 *	1.46; 5.02
3	154 94.48%	202 87.07%	5.907	0.0151	2.54	1.17; 5.51
4 #	108 66.26%	66 28.45%	55.530	0.0000	4.94 *	3.21; 7.61
5 #	95 58.28%	79 34.05%	22.806	0.0000	2.71	1.79; 4.09
6	27 16.56%	24 10.34%	3.293	0.0695	1.72	0.95; 3.11
7	45 27.61%	36 15.52%	8.584	0.0033	2.08	1.27; 3.40
8 #	23 14.11%	11 4.76%	10.592	0.0011	3.29 *	1.55; 6.95
9 #	95 58.28%	89 38.36%	15.267	0.0001	2.24 *	1.49; 3.38
10 #	90 55.21%	85 36.64%	13.389	0.0003	2.13 *	1.41; 3.21
11 #	32 19.63%	11 4.74%	21.882	0.0000	4.91 *	2.39; 10.06
12 #	109 66.87%	66 28.45%	57.278	0.0000	5.08 *	3.29; 7.83
13 #	86 52.76%	11 4.74%	119.156	0.0000	22.44 *	11.38; 44.25
14 #	160 98.16%	201 86.64%	16.155	0.0001	8.23 *	2.47; 27.39
15	149 91.41%	192 82.76%	6.07	0.0137	2.21	1.16; 4.23
16 #	130 79.75%	147 63.36%	12.280	0.0005	2.28 *	1.43; 3.63
17 #	98 60.12%	69 29.74%	36.21	0.0000	3.56 *	2.34; 5.43
18	16 9.82%	19 8.19%	0.313	0.5755	1.22	0.60; 2.45
19	156 95.71%	223 96.12%	0.042	0.8367	0.90	0.33; 2.47
20	97 59.51%	107 46.12%	6.872	0.0087	1.71	1.14; 2.58
21 #	134 82.21%	156 67.24%	10.989	0.0009	2.25 *	1.38; 3.66
22	155 95.09%	214 92.24%	1.265	0.2606	1.63	0.69; 3.84
23 #	57 34.97%	46 19.83%	11.38	0.0007	2.17 *	1.38; 3.43
24	132 80.98%	167 71.98%	4.214	0.0401	1.66	1.02; 2.69
25	52 31.90%	76 32.76%	0.032	0.8578	0.96	0.63; 1.48
26	85 52.15%	95 40.95%	4.840	0.0277	1.57	1.05; 2.35
27 #	70 42.94%	40 17.24%	31.479	0.0000	3.61	2.28; 5.73
28	110 67.48%	164 70.69%	0.462	0.4963	0.86	0.56; 1.33
29	37 22.70%	55 23.71%	0.054	0.8155	0.94	0.59; 1.52
30	19 11.66%	26 11.21%	0.019	0.8898	1.04	0.56; 1.96
31	15 9.20%	16 6.90%	0.704	0.4014	1.37	0.66; 2.85
32 #	109 66.87%	147 63.36%	0.517	0.4722	1.16	0.77; 1.78
33 #	126 77.30%	166 71.55%	1.641	0.2001	1.35	0.85; 2.15
			Z	*p*-Value		
Total Metylation Level (%) *	56.70 ± 18.82	42.85 ± 16.25	7.125	<0.00001		
Number of methylation islands *	18.71 ± 6.21	14.14 ± 5.36	7.125	<0.00001		

χ^2^ (*p*)—chi-square test (significance level); #—differences in the level of methylation at the limit of statistical significance; n—number of subjects. # Bonferroni correction was applied, and the *p*-value was lowered to 0.0015 (*p* = 0.05/33 (number of statistical tests performed)). *p*-value of statistical significance in Mann–Whitney U-test; n, number of subjects; M ± SD, mean ± standard deviation; * differences which are statistically significant (*p* < 0.005).

**Table 2 ijms-24-08931-t002:** The results of the NEO Five-Factor Inventory for athletes and controls.

NEO FFI	Athletes n = 163	Controls n = 232	Z (*p*-Value)
Neuroticism/scale	4.81 ± 2.21	4.61 ± 1.90	0.659 (0.5097)
Extraversion/scale *	6.84 ± 2.01	6.36 ± 2.00	2.154 (0.0312 *)
Openness/scale	4.86 ± 2.50	4.53± 1.63	1.554 (0.1200)
Agreeability/scale	5.94 ± 3.74	5.66 ± 2.07	0.416 (0.6768)
Conscientiousness/scale *	7.19 ± 2.09	5.88 ± 2.12	5.853 (<0.00001 *)

*p*-value of statistical significance in Mann–Whitney U-test; n, number of subjects; M ± SD, mean ± standard deviation; * differences which are statistically significant (*p* < 0.05).

**Table 3 ijms-24-08931-t003:** Pearson’s linear correlation between total methylation level, number of methylation islands and NEO-FFI in a group of athletes and controls.

	Neuroticism Scale	Extraversion Scale	Openness Scale	Agreeability Scale	Conscientiousness Scale
Total Methylation Level (%) Athletes	r = −0.069 (*p* = 0.387)	r = 0.174 (*p* = 0.027)	r = 0.106 (*p* = 0.182)	r = 0.158 (*p* = 0.044)	r = 0.128 (*p* = 0.105)
Total Methylation Level (%) Controls	r = 0.026 (*p* = 0.690)	r = −0.118 (*p* = 0.073)	r = −0.043 (*p* = 0.511)	r = 0.087 (*p* = 0.187)	r = 0.003 (*p* = 0.964)

r—Pearson’s linear correlation; *p*-value of statistical significance.

**Table 4 ijms-24-08931-t004:** Primary statistics of analyzed groups.

	Athletes	Controls
N	163	232
Age M(SD)	23.31 (5.65)	22.53 (4.86)
Man/Woman	85%/15%	90%/10%

**Table 5 ijms-24-08931-t005:** PCR reaction conditions for the amplification of a 447 bp fragment encompassing 33 CpG sites in the promoter of the *DAT1* gene.

Number of Cycles	PCR Step	Temperature	Time
1	Initial Denaturation Denaturation	94 °C 94 °C	5:00
			0:25
35	Annealing Elongation	61 °C 72 °C	0:25
			0:25
1	Final elongation	72 °C	5:00

## Data Availability

Not applicable.

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
