# Peer review of "Epigenetic Analysis of the Dopamine Transporter Gene DAT1 with a Focus on Personality Traits in Athletes"

_ijms, 2023, doi:10.3390/ijms24108931_

Round 1

Reviewer 1 Report

This is an interesting manuscript on an important topic. The paper is generally well written and I only have some minor suggestions that could improve the readers' understanding.

At page 1, line 42 I suggest adding more example about what factors can influence human phenotype. See: Cimino, S., et al.,  DNA methylation at the DAT promoter and risk for psychopathology: Intergenerational transmission between school-age youths and their parents in a community sample.

Cimino, S., et al., Children’s dat1 polymorphism moderates the relationship between parents’ psychological profiles, children’s dat methylation, and their emotional/behavioral functioning in a normative sample.

At page 2, line 78, add relevant references.

In the introduction I suggest adding at least a brief, tentative description of a model showing how psychological factors are associated with epigenetic modifications, also considering exercise.

Methods and procedure paragraph should be moved after the introduction section.

English is understandable and fluent.

Author Response

Dear Reviewer,

Thank you very much for your review and valuable comments, which we apologize for and for showing shortcomings and nonsense in our Manuscript. We analyzed all the comments and replied to each, indicating where and how the corrections in the Manuscript were made, indicating the line and page. Below are the point-by-point answers.

With respect

At page 1, line 42 I suggest adding more example about what factors can influence human phenotype. See: Cimino, S., et al.,  DNA methylation at the DAT promoter and risk for psychopathology: Intergenerational transmission between school-age youths and their parents in a community sample.

Cimino, S., et al., Children’s dat1 polymorphism moderates the relationship between parents’ psychological profiles, children’s dat methylation, and their emotional/behavioral functioning in a normative sample.

Thank you for this suggestion, We added more examples. Page 2 Line 51-52

At page 2, line 78, add relevant references.

Thank you for this comment, a more appropriate reference has been added to the text Page 3 Line 102-103

In the introduction I suggest adding at least a brief, tentative description of a model showing how psychological factors are associated with epigenetic modifications, also considering exercise.

Thank you for this suggestion. The excerpt suggested by the Reviewer has been added in the introduction. Page 2 line 79-86

Methods and procedure paragraph should be moved after the introduction section.

We used the Microsoft Word template from the instructions for authors and there it is advisable to present the results in the second chapter.

Reviewer 2 Report

This manuscript investigated the association between methylation in the promoter region of dopamine transporter DAT1 gene and personality traits that represent important determinants of success in sports. This study is very interesting and brings novelty to the new research field of sports epigenomics. Besides, the manuscript is generally well designed and written, methods are appropriate and adequate experiments have been performed. However, there are several concerns that need to be clarified and modified before publication.

First, the title needs to be more informative and descriptive, so please change it. The current form of the title looks like a mix of key words without any explanation.

Conclusions of the study need to be added in the Abstract.

Results should be explained more in detail. Every table should be explained in the text separately. For example, in the last paragraph (lines 132-134), it should be added that no correlation was shown for the control group.

Lines 172, 178 – it should be Mirzaei et al and Steca et al.

Lines 191-193 – it should be concluded that the type of sport has an impact.

Conclusions should be more extensive. The main results should be summarized and then concluded.

Language is fine.

Author Response

Dear Reviewer,

Thank you very much for your review and valuable comments, which we apologize for and for showing shortcomings and nonsense in our Manuscript. We analyzed all the comments and replied to each, indicating where and how the corrections in the Manuscript were made, indicating the line and page.

Below are the point-by-point answers.

With respect

First, the title needs to be more informative and descriptive, so please change it. The current form of the title looks like a mix of key words without any explanation.

Thank you for those suggestions. This is of course a mistake. The title has been corrected.

Conclusions of the study need to be added in the Abstract.

Thank you for this great suggestions. This has been added to the text.. Page 1 line 41-47

Results should be explained more in detail. Every table should be explained in the text separately. For example, in the last paragraph (lines 132-134), it should be added that no correlation was shown for the control group.

Thank you for this comment, The description of Tables 1,2 and 3 in the text is detailed.

Lines 136-140, 150-152, 155-160

Lines 172, 178 – it should be Mirzaei et al and Steca et al.

Thank you for this suggestion. This has been corrected in the text. page line 198 and 204

Lines 191-193 – it should be concluded that the type of sport has an impact.

Thank you for this great suggestions. This has been added to the text. Line 219-220

Conclusions should be more extensive. The main results should be summarized and then concluded.

Thank you for this suggestion. This has been added in text. Page line 327-330

Round 2

Reviewer 2 Report

The manuscript is improved.

Please rephrase the sentence in the line 225 (delete "as you can see").

No major issues detected.